# Evaluation of the Kinetics of Pancreatic Stone Protein as a Predictor of Ventilator-Associated Pneumonia

**DOI:** 10.3390/biomedicines11102676

**Published:** 2023-09-29

**Authors:** Adrian Ceccato, Marta Camprubí-Rimblas, Lieuwe D. J. Bos, Pedro Povoa, Ignacio Martin-Loeches, Carles Forné, Aina Areny-Balagueró, Elena Campaña-Duel, Luis Morales-Quinteros, Sara Quero, Paula Ramirez, Mariano Esperatti, Antoni Torres, Lluis Blanch, Antonio Artigas

**Affiliations:** 1Critical Care Center, Institut d’Investigació i Innovació Parc Taulí I3PT-CERCA, Hospital Universitari Parc Taulí, Univeristat Autonoma de Barcelona, 08208 Sabadell, Spain; mcamprubi@tauli.cat (M.C.-R.); aareny@tauli.cat (A.A.-B.); ecampanad@tauli.cat (E.C.-D.); luchomq2077@gmail.com (L.M.-Q.); squero@tauli.cat (S.Q.); lblanch@tauli.cat (L.B.); 2CIBER of Respiratory Diseases (CIBERES), Institute of Health Carlos III, 28029 Madrid, Spain; drmartinloeches@gmail.com (I.M.-L.); atorres@clinic.cat (A.T.); 3Intensive Care Unit, Hospital Universitari Sagrat Cor, Grupo Quironsalud, 08029 Barcelona, Spain; 4Intensive Care & Laboratory of Experimental Intensive Care and Anesthesiology (LEICA), Amsterdam UMC Location AMC, University of Amsterdam, 1105 AZ Amsterdam, The Netherlands; l.d.bos@amsterdamumc.nl; 5Department of Critical Care Medicine, Hospital de São Francisco Xavier, CHLO, 1449-005 Lisbon, Portugal; pedro.povoa@nms.unl.pt; 6Nova Medical School, New University of Lisbon, 1169-056 Lisbon, Portugal; 7Center for Clinical Epidemiology and Research Unit of Clinical Epidemiology, OUH Odense University Hospital, 5000 Odense, Denmark; 8Department of Intensive Care Medicine, Multidisciplinary Intensive Care Research Organization (MICRO), St. James Hospital, D08 NHY1 Dublin, Ireland; 9Department of Pneumology, Hospital Clinic of Barcelona—August Pi i Sunyer Biomedical Research Institute (IDIBAPS), University of Barcelona, 08036 Barcelona, Spain; 10Heorfy Consulting, 25007 Lleida, Spain; carles.forne@heorfy.com; 11Department of Basic Medical Sciences, University of Lleida, 25198 Lleida, Spain; 12Servei de Medicina Intensiva, Hospital de la Santa Creu y Sant Pau, 08025 Barcelona, Spain; 13Servicio de Medicina Intensiva, Hospital Universitario y Politécnico la Fe, 46026 Valencia, Spain; ramirez_pau@gva.es; 14Escuela Superior de Medicina, Universidad Nacional de Mar del Plata, Mar del Plata B7602AYL, Argentina; marianoesperatti@gmail.com; 15Unidad de Cuidados Intensivos, Hospital Privado de Comunidad, Mar del Plata B7602AYL, Argentina

**Keywords:** ventilator-associated pneumonia, pancreatic stone protein, biomarkers, prediction

## Abstract

BACKGROUND: Ventilator-associated pneumonia (VAP) is a severe condition. Early and adequate antibiotic treatment is the most important strategy for improving prognosis. Pancreatic Stone Protein (PSP) has been described as a biomarker that increases values 3–4 days before the clinical diagnosis of nosocomial sepsis in different clinical settings. We hypothesized that serial measures of PSP and its kinetics allow for an early diagnosis of VAP. METHODS: The BioVAP study was a prospective observational study designed to evaluate the role of biomarker dynamics in the diagnosis of VAP. To determine the association between repeatedly measured PSP and the risk of VAP, we used joint models for longitudinal and time-to-event data. RESULTS: Of 209 patients, 43 (20.6%) patients developed VAP, with a median time of 4 days. Multivariate joint models with PSP, CRP, and PCT did not show an association between biomarkers and VAP for the daily absolute value, with a hazard ratio (HR) for PSP of 1.01 (95% credible interval: 0.97 to 1.05), for CRP of 1.00 (0.83 to 1.22), and for PCT of 0.95 (0.82 to 1.08). The daily change of biomarkers provided similar results, with an HR for PSP of 1.15 (0.94 to 1.41), for CRP of 0.76 (0.35 to 1.58), and for PCT of 0.77 (0.40 to 1.45). CONCLUSION: Neither absolute PSP values nor PSP kinetics alone nor in combination with other biomarkers were useful in improving the prediction diagnosis accuracy in patients with VAP. Clinical Trial Registration: Registered retrospectively on August 3rd, 2012. NCT02078999.

## 1. Introduction

Ventilator-associated pneumonia (VAP) is a severe condition in patients receiving invasive mechanical ventilation and is a common nosocomial infection [1]. Its incidence is variable, with approximately 20–30% of patients receiving invasive mechanical ventilation developing VAP. The mortality of patients with VAP is high and can be more than 60% in patients with VAP caused by multi-resistant microorganisms. Early and appropriate antibiotic treatment is the most important strategy for improving clinical outcomes, reducing the risk of death [2]. However, early diagnosis is not always achieved due to the lack of a gold standard for diagnosis, the presence of comorbidities, and delays in culture results [2,3]. VAP is a preventable condition, and bundled interventions have been shown to improve the prevalence of this condition [4]. Nevertheless, during the COVID-19 pandemic, the incidence of VAP increased [5], as did the associated mortality, highlighting the need for new tools aimed at improving the care of patients under invasive mechanical ventilation [6].

Using a single time point, no biomarker has shown good predictive accuracy in improving the diagnosis of VAP or in discriminating with other infectious conditions such as tracheobronchitis [7]. C-reactive protein (CRP) and procalcitonin (PCT) are the most studied biomarkers. In the Biomarkers in the diagnosis and management of VAP study (BioVAP), the highest ratio, delta, and kinetics in the serial measurement of CRP were associated with the development of VAP [8]. Other biomarkers, including cytokines, chemokines, amylase in respiratory specimens [9], and alveolar proteins, have been proposed for the diagnosis of VAP [10,11], but none have shown sufficient clinical utility for widespread implementation.

Pancreatic Stone Protein (PSP) is a biomarker secreted by the pancreas [12] and has multiple functions, including adhesion and signaling receptors in homeostasis and innate immunity, and it is critical for inflammatory responses and leukocyte and platelet trafficking [13,14]. PSP has been proposed as a biomarker that is potentially useful in predicting nosocomial sepsis, as it increases the 3–4 days before clinical diagnosis in different settings of patients [15]. These increases in PSP levels may allow for early identification and improve the clinical management of patients with sepsis. Of no less importance, PSP can be accurately measured easily and quickly at the point of care.

Given the poor predictive ability of VAP biomarkers, we investigated the potential of PSP to establish the diagnosis of VAP. We hypothesized that serial measurements of PSP and its kinetics would allow for the early diagnosis of VAP. We aimed to evaluate the diagnostic utility of PSP levels or longitudinal trajectories for VAP diagnosis.

## 2. Methods

This is a secondary analysis using plasma samples from the BioVAP study. The BioVAP study was a prospective, European multicenter, observational study designed to evaluate the additional information biomarkers can bring in the clinical decision-making process of VAP at the bedside. Further details can be found elsewhere, and the study was previously registered (NCT02078999) [8]. This study is reported following STROBE guidelines [16]. The institution’s Ethical Review Board and each local institutional committee approved the study (Comité de Ética de Investigación con medicamentos del Parc Taulí 2008/524), which was performed following the ethical standards laid down in the 1964 Declaration of Helsinki and its later amendments. Written informed consent was obtained from all patients or their legally authorized surrogates.

### 2.1. Inclusion and Exclusion Criteria

The inclusion criteria were: (1) Patients admitted to the ICU with an expected duration of mechanical ventilation of more than 3 days; (2) Not receiving antibiotics for more than 24 h before ICU admission. The exclusion criteria were: Patients younger than 18 years, pregnancy and lactation, fulminant liver failure, pancreatitis, patients with a diagnosis of disseminated cancer, and expected to die or withdraw treatment within 72 h of enrollment.

### 2.2. Measurement of PSP

Plasma samples had been collected: 4 mL of blood was collected in vacuum tubes and centrifuged at 3000 rpm for 10 min, and plasma samples were obtained and frozen at −80 °C in a biobank. To analyze PSP, plasma samples were thawed, leaving them for 30 min at room temperature. PSP levels were determined using the IVD CAPSULE PSP Plasma on the CE-marked point-of-care abioSCOPE^®^ device (Abionic SA, Geneva, Switzerland), according to the manufacturer’s protocol. The results in plasma are directly comparable to the results in whole blood measured on the abioSCOPE^®^ device.

### 2.3. Definitions

Infection was defined as a pathologic process caused by the invasion of tissue, fluid, or a body cavity by a pathogenic or potentially pathogenic microorganism and/or clinically suspected infection, plus the prescription of antimicrobial therapy. Community-acquired infection, either of pulmonary or extra-pulmonary origin, was defined as the onset of infection before hospital admission or not present at admission but becoming evident in the first 48 h. All infections diagnosed after 48 h of the hospital stay were classified as hospital-acquired.

The clinical diagnosis of VAP was defined as a new and persistent radiographic pulmonary infiltrate plus at least two of the following criteria: (a) temperature >38 °C or <36 °C; (b) white cell count (WCC) >10 or <4 × 10^3^/mm^3^; and (c) purulent tracheal aspirate. The chest X-rays were reviewed either by the attending physicians or a radiologist. In case of disagreement, a third physician was asked to interpret the chest X-ray. It was mandatory to perform a QTA, a bronchoscopic or non-bronchoscopic BAL, and at least two separate blood cultures. The thresholds used for the diagnosis of pneumonia were ≥10^5^ colony-forming units (CFU)/mL on a QTA and ≥10^4^ CFU/mL on a BAL.

### 2.4. Outcomes

The primary research question was to evaluate the association of PSP with the diagnosis of the first VAP episode in patients admitted to the ICU under invasive mechanical ventilation. Non-VAP secondary infections or ventilator-associated tracheobronchitis were not evaluated since the study was not designed to this end. During the study, the patients were followed up until they developed VAP, were extubated, or died.

### 2.5. Statistical Analysis

We used mixed-effects models to explore the population longitudinal trends of PSP from intubation and before/after VAP onset. To determine the association between repeatedly measured PSP, CRP, and PCT and the risk of incident VAP, we used joint models for longitudinal and time-to-event data [17]. These models account for the correlations in the repeated measurements of individuals and their endogenous nature, combining mixed-effects models for repeated measurements of biomarkers with a time-to-event relative risk model for the time-to-event data. The timescale in the joint models was days of follow-up, which started at the intubation. The VAP risk submodel was adjusted for baseline infection, and the longitudinal submodels included natural cubic splines for the follow-up time, which was also adjusted for baseline infection and additionally for the APACHE II score and CRP level at ICU admission. Beta distribution was used to fit PSP trajectories, with boundaries at 20 and 600, normalizing to a 0–1 range as (PSP − 20)/(600 − 20); and Gaussian distribution was used to fit trajectories in the log-scale of CRP and PCT. Several models were fitted, including combinations of biomarkers and several functional forms in the hazard of VAP: model 1, the value, daily change, and cumulative area of PSP-normalized; model 2, the value of PSP-normalized, log(CRP), and log(PCT); model 3, the daily change of PSP-normalized, log(CRP), and log(PCT); and model 4, both the value and daily change of PSP-normalized, log(CRP), and log(PCT). The time-dependent area under the curve (AUC) methodology, as adapted for joint models, was used to determine the longitudinal markers’ prospective accuracy [18].

We used a Bayesian approach to fit the joint models. Wide proper prior distributions were used—in particular, those defaulted by the R function JMbayes2::jm. Posterior distributions were approximated by means of three MCMC chains with 600,000 iterations, 60,000 of which were used for the burn-in period. The chains were thinned by only storing every 100th iteration to reduce autocorrelation in the saved samples. Trace plots of the simulated values of the three chains appeared to overlap one another, indicating stabilization. The convergence of the chains to the posterior distribution was assessed through the potential scale reduction factor, $\hat{R}$ (all of them were near one, indicating that the simulated process had reached the posterior distribution).

R version 4.2.0 was used for all analyses [19]. The mixed_model function of the GLMMadaptive package [20] was used to fit the mixed-effects models; the jm function of the JMbayes2 package [21] was used to fit the joint models; and the results were visualized using the ggplot2 package [22].

## 3. Results

Of 211 patients consecutively included in the BioVAP study, 209 were analyzed (Figure 1). Baseline characteristics are described in Table 1, overall, and by the VAP diagnostic status. Of 209 patients, 73 (34.9%) patients presented an infection at admission to the ICU (pulmonary N = 44 or non-pulmonary n = 29). The median age was 63 years, and the most common comorbid conditions were heart failure, diabetes, and chronic obstructive pulmonary disease (COPD). In 158 (75.6%) patients, the main cause of hospital admission was medical and the main cause of mechanical ventilation was respiratory failure (40.7%).

Eighty-nine patients (42.6%) presented a nosocomial infection during their ICU stay. Forty-three (20.6%) presented a VAP episode. The median time to VAP diagnosis was 4 days (P25-P75: 3, 6.5), that to death was 8 days (3, 17), and that to weaning was 7 days (4, 11).

A total of 639 measures of PSP, 1295 of CRP, and 1091 of PCT were available for analysis. Longitudinal PSP, CRP, and PCT trends are shown in Figure 2, according to the presence of infection at admission and VAP development. Raw and adjusted values of PSP before and after VAP are detailed in Table 2.

The impact of the PSP, CRP, and PCT on the risk of VAP was evaluated by means of joint modeling. Posterior summaries of all models are detailed in the Appendix A. None of the PSP functional forms analyzed (value, daily change, and cumulative area) showed a relevant impact on predicting the risk of VAP during the follow-up period (model 1; see Appendix A). Multivariate joint models with PSP, CRP, and PCT also did not show a relevant impact of any of the biomarkers, neither with the current value nor with the daily change or both (models 2–4; see Appendix A). Only infection at admission was associated with a decreased hazard of VAP, obtaining a very stable and robust estimation from all models. The presence of infection at admission was shown to be associated with higher levels of PCT, and greater severity measured with the APACHE II was shown to be associated with higher levels of PSP and PCT (see Appendix A). We performed a sensitivity analysis with a non-infected population, and similar results were obtained for PSP (Appendix A). Table 3 shows the hazard ratios provided by the models fitted. All models showed a very similar calibration, assessed by means of the Brier score, with values from 0.208 for model 4 to 0.213 for model 1. As expected given the results provided by the models, the discrimination capability was very low for all models, with AUC estimates under 0.65 for any starting and horizon time points.

We explored the role of death and extubation as competing risks. The cumulative incidences estimated with the Kaplan–Meier method or the competing risks with the cause-specific hazards approach are very similar until day 7, by which time more than 80% of VAP were already diagnosed (see Appendix A).

The models fitted can be used to provide individualized predictions for the risk of VAP, utilizing new information as time progresses. Figure 3 shows the dynamic predictions (posterior mean and 95% credible band at days 4, 6, and 10 after intubation, provided by model 4) for four patients with or without infection at admission and with or without VAP at the end of follow-up: 1–112 (infection at admission and no VAP), 4–10010 (no infection and no VAP), 4–10040 (no infection and VAP), 7–21 (infection and VAP). As expected, the risk of VAP is low and decreases as the days progress. It is worth noting the tightening of the bands corresponding to predictions from later days, with a higher number of measurements, in contrast to the wide bands at the beginning of the follow-up. Patients with infection at admission seem to have a lower risk of VAP, showing the risk-reduction effect provided by the models. However, we must also consider the longitudinal biomarkers. Thus, the risk of VAP is higher for patient 7–21, who has a stable, very low CRP and PCT and a stable, high PSP, than for patient 1–112, who has higher levels of CRP and PCT throughout the follow-up. Both patients have an infection at admission.

Trends fitted by means of mixed-effects models. CRP, C-reactive protein; PCT, procalcitonin; PSP, pancreatic stone protein; VAP, ventilator-associated pneumonia.

The posterior mean and 95% credible band of the longitudinal biomarkers trend (blue) and the probability of VAP diagnosis (red) for selected patients with the IDs 1–112 (infected at ICU admission and without VAP diagnosis), 4–10010 (non-infected at ICU admission and without VAP diagnosis), 4–10040 (non-infected at ICU admission and with VAP diagnosis), and 7–21 (infected at ICU admission and with VAP diagnosis) are shown. The value of the probability at the upper right of each graphic is the subsequent posterior mean of VAP risk at 14 days from intubation, conditioned to VAP having not been diagnosed at 4, 6, or 10 days from intubation.

## 4. Discussion

Our main finding was that PSP values were not useful in predicting VAP development. Additionally, when PSP was considered while combined with other biomarkers, the diagnosis performance did not improve for VAP. We analyzed a well-collected cohort of patients who received invasive mechanical ventilation and PSP measures performed in collected frozen samples of plasma.

A potential benefit and the main objective of this research was the availability of PSP measured at the bedside, with fast-track results after six minutes. This characteristic added to the predictive performance in nosocomial sepsis, making it an attractive diagnostic tool for critically ill patients. Pugin et al. [15] found in a multicenter study with 243 patients that PSP has a moderate accuracy (AUC 0.75), similar to other biomarkers such as PCT or CRP, in diagnosing nosocomial sepsis. A significant increase in PSP values from baseline three days before sepsis diagnosis was observed [15]. Also, PSP was evaluated at the emergency department, reaching a good accuracy (AUC 0.84) in evaluating the prognosis of sepsis in critically ill patients during the first 48 h [23]. In a meta-analysis [24] including five observational studies, the accuracy of diagnosis of sepsis was slightly higher, with an AUC of 0.81. The cutoff chosen for PSP was 44 ng/mL. In our study, the means of the posterior distributions of all functional forms (value, daily change, and/or cumulative area) for PSP showed no relevant impact on the early prediction of VAP. Moreover, high variability in the measures of PSP was observed in our study, leading to highly credible intervals and reducing the accuracy of the biomarker. We were not able to analyze the correlation of the bacterial load and PSP level determination with a higher response of PSP; however, in this study, we analyzed the association of other markers of systemic inflammation joining with PSP, and the accuracy was not improved.

Interestingly, there is no evidence of the expression of PSP or its gen REG1B in the lung according to publicly available datasets [25,26]. Thus, PSP levels may rise only as a consequence of extra lung sepsis or VAP with a non-compartmentalized immune response. VAP has shown a different expression of a biomarkers profile when comparing measures in bronchoalveolar lavage or serum, probably due to the compartmentalization of the immune response [27,28].

PSP was evaluated in patients with VAP as an outcome predictor by Boeck et al. [29], showing elevated values in non-survivor patients, mainly after 7 days of VAP onset. Also, higher PSP levels were related to the presence of organ dysfunctions, but the impact of longitudinal changes or predictive diagnosis was not evaluated in this study.

The diagnosis of VAP remains a major challenge faced by clinicians daily. Ideally, a biomarker should be able to predict an early and accurate diagnosis of VAP so that we can provide early and appropriate treatment to reduce the risk of death associated with this condition [30]. However, no single biomarker measurement has shown sufficient diagnostic accuracy, and none is recommended in clinical guidelines [2,31]. Despite the lack of evidence and recommendations for the use of biomarkers in the diagnosis of VAP, these tools are widely used by clinicians [32]. A single level of CRP and PCT has been evaluated to differentiate ventilator-associated tracheobronchitis from VAP. Although significant differences were found, neither biomarker allows for an adequate diagnosis due to a significant overlap between both conditions [7]. Previously, several biomarkers were evaluated in this cohort of patients, with the ratio of CRP and slope being the most accurate method for diagnosis [8]. The slope of CRP had an adjusted OR of 1.62 (95% CI 1.20 to 2.18), and the highest CRP ratio had an aOR of 1.20 (95% CI 1.06 to 1.36). Neither PCT, pro-adrenomedullin, leukocytes, nor temperature showed significant accuracy in predicting VAP in this analysis. Although a significant association between a biomarker and a disease is required, this does not mean that the biomarker is accurate enough to discriminate between patients with and without the disease [33]. The current study used a different methodology and selected population than previous analyses, and it is worth mentioning that we did not analyze CRP or PCT alone in this study, as this was not aimed to replicate the previous analyses. Models include CRP and PCT as covariates associated with PSP.

Joint modeling allows us to analyze the longitudinal submodel with serial measures of endogenous covariates in a mixed-effects model and the time-to-event or survival submodel by Cox regression. Joint models were developed to account for the special features of endogenous covariates, for which standard time-varying Cox models are not appropriate [17]. Rue et al. [34] pointed out that joint models of longitudinal and survival data are the most appropriate approach for assessing the effect of potential biomarkers or risk factors in patients admitted to ICUs and for dynamically updating the patient prognosis. This method was also applied to other ICU populations and showed improved prediction with the use of longitudinal data [35]. In addition, the use of a Bayesian approach allowed us to explore the impact of PSP, along with other biomarkers, on the risk of VAP through multiple functional forms. Models were adjusted for variables that could have an impact on measures or outcomes. We used time-to-event data and censored cases if death or extubation occurred. Although this method allows for competing risks, we prefer censored cases because of the small n for each event. We explored the role of death and extubation as competing risks, and the cumulative incidences estimated with the cause-specific hazards approach are very similar until day 7, by which time more than 80% of VAP were already diagnosed. Therefore, even though the censoring due to the competing risks was informative, it would hardly affect our estimates.

This study has strengths and limitations that should be acknowledged. The strengths included the evaluation of a well-characterized cohort of patients aimed at evaluating the role of biomarkers in the diagnosis of VAP and the use of joint models. The main limitations of the study were that, despite an important number of patients recruited, the number of events (VAP) limits the capacity of the model, and frozen samples were not available for all days for all patients. We included patients with an infection at admission despite these patients being excluded in the prior analysis of this cohort. We think that including the overall cohort allows us to increase the number of patients and show a real-life aspect. Moreover, a sensitivity analysis with non-infected patients at admission was performed, showing similar results. Patients with infection at admission had a low risk of VAP in our model, probably because patients with infection at admission are under antibiotic treatment; however, we do not have enough data to clarify this point, and the study was not designed with this purpose.

## 5. Conclusions

In our cohort of patients, neither absolute PSP levels nor PSP kinetics alone or in combination with other biomarkers were found to be useful in improving the accuracy of predicting the diagnosis of patients with VAP. Thus, PSP measurements have limited utility in clinical practice. Further studies are needed to determine whether PSP could be valid in predicting the development of non-VAP nosocomial sepsis.

## Figures and Tables

**Figure 1 biomedicines-11-02676-f001:**
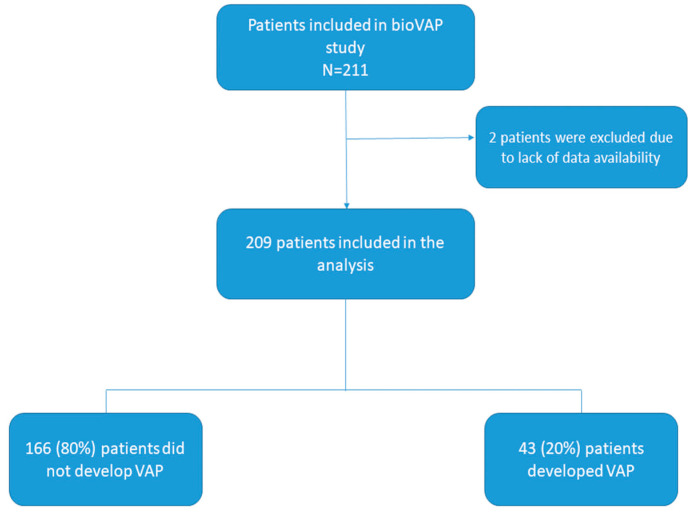
Flow chart.

**Figure 2 biomedicines-11-02676-f002:**
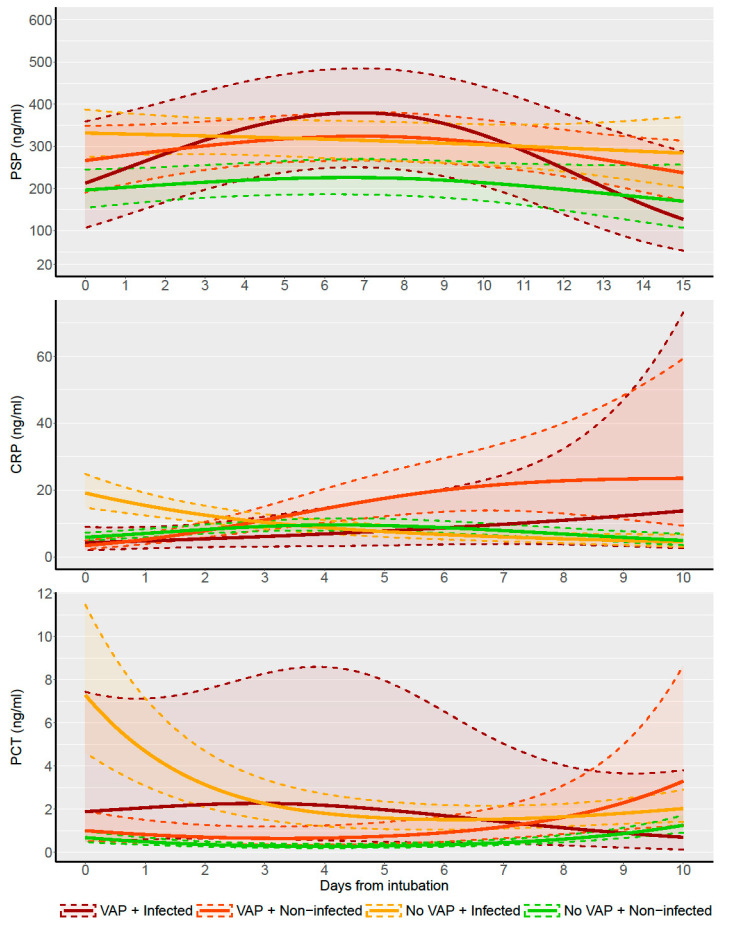
Longitudinal trends of PSP, CRP, and PCT.

**Figure 3 biomedicines-11-02676-f003:**
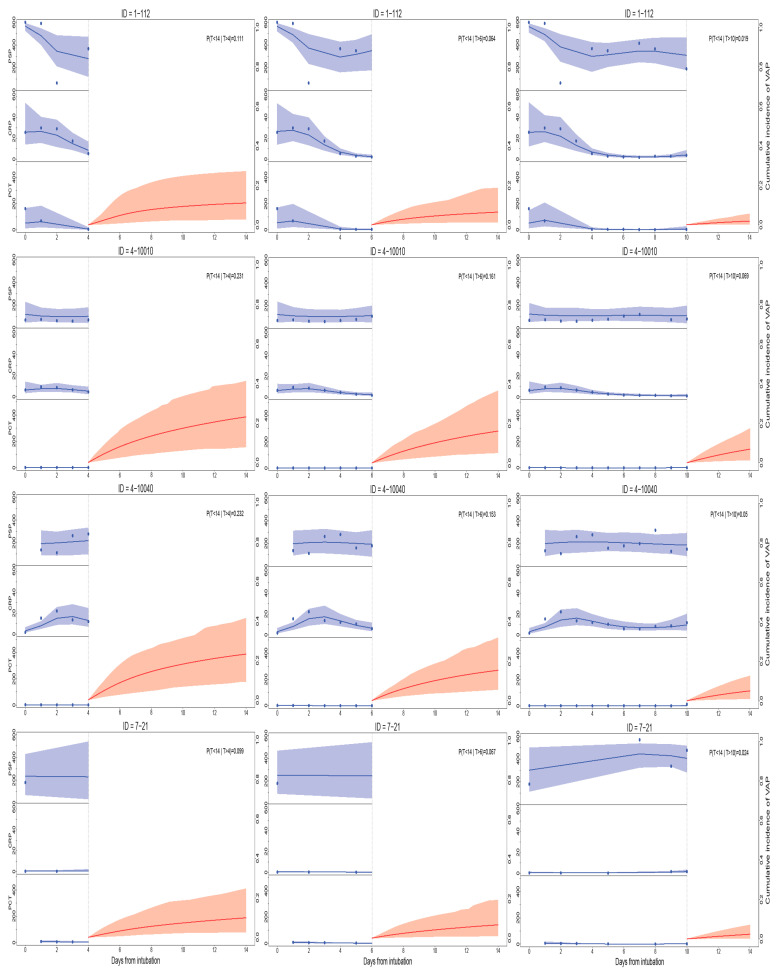
Dynamic predictions from model 4 for four patients.

**Table 1 biomedicines-11-02676-t001:** Baseline characteristics, longitudinal measures of biomarkers, and outcomes.

Characteristic	Overall (N = 209)	Non-VAP (N = 166)	VAP (N = 43)
Age, years	63 (48, 74)	64.5 (50.2, 75)	59 (41, 66)
Female	71 (34%)	60 (36.1%)	11 (25.6%)
Main cause of hospital admission			
Medical	158 (75.6%)	127 (76.5%)	31 (72.1%)
Elective surgery	30 (14.4%)	20 (12.0%)	10 (23.3%)
Trauma	3 (1.44%)	3 (1.81%)	0 (0.00%)
Emergency surgery	18 (8.61%)	16 (9.64%)	2 (4.65%)
Infection at ICU admission			
*Non-infected*	136 (65.1%)	101 (60.8%)	35 (81.4%)
*Infected*	73 (34.9%)	65 (39.2%)	8 (18.6%)
Site of infection at ICU admission			
Pulmonary infection	44 (60.3%)	38 (22.9%)	6 (14.0%)
Non-pulmonary infection	29 (39.7%)	27 (16.3%)	2 (4.65%)
Corticosteroid	8 (3.83%)	6 (3.61%)	2 (4.65%)
COPD	31 (14.8%)	21 (12.7%)	10 (23.3%)
Diabetes	31 (14.8%)	26 (15.7%)	5 (11.6%)
Immunosuppression	14 (6.70%)	10 (6.02%)	4 (9.30%)
Heart failure	34 (16.3%)	30 (18.1%)	4 (9.30%)
Renal failure	19 (9.09%)	14 (8.43%)	5 (11.6%)
HIV	5 (2.39%)	4 (2.41%)	1 (2.33%)
Cause of mechanical ventilation			
Respiratory failure	85 (40.7%)	73 (44.0%)	12 (27.9%)
Shock	33 (15.8%)	25 (15.1%)	8 (18.6%)
Coma	85 (40.7%)	64 (38.6%)	21 (48.8%)
Other	6 (2.87%)	4 (2.41%)	2 (4.65%)
Apache II at ICU admission	23 (17, 29)	22 (17, 29.8)	24 (19.5, 28.5)
SAPS II at ICU admission	50.3 (19)	49.4 (19.4)	53.6 (17.0)
CRP, mg/dL			
At admission	5.38 (1.59, 12.8)	6.36 (1.84, 17.1)	4.3 (1.1, 7.1)
Day 2	12.4 (7.89, 19.9)	13.1 (8.55, 19.9)	9.70 (5.9, 16.3)
Day 5	10.2 (4.65, 16.5)	9.57 (4.82, 15.7)	13.7 (4.54, 25.9)
Day 7	9.05 (3.95, 14.4)	9.05 (3.8, 13.7)	12.4 (6.11, 21.8)
PCT, ng/mL			
At admission	0.83 (0.21, 6.88)	0.74 (0.20, 7.58)	1.19 (0.24, 2.44)
Day 2	0.88 (0.2, 4.84)	0.76 (0.2, 5.05)	1.02 (0.38, 4.49)
Day 5	0.31 (0.15, 1.75)	0.31 (0.14, 1.63)	0.68 (0.16, 2.67)
Day 7	0.38 (0.13, 1.21)	0.38 (0.13, 1.25)	0.13 (0.13, 1.09)
PSP, ng/mL			
At admission	132 (68, 352)	110 (54, 372)	161 (129, 202)
Day 2	136 (67, 296)	134 (60, 306)	165 (77.5, 276)
Day 5	208 (93, 370)	195 (79.5, 331)	291 (151, 411)
Day 7	146 (99, 428)	145 (90.5, 396)	184 (124, 534)
PaO_2_/FiO_2_ at admission	198 (135, 311)	192 (128, 317)	233 (177, 287)
Hospital LOS, days	26 (16, 47)	26 (17, 46.8)	27 (16, 48)
In-ICU mortality	47 (22.5%)	34 (20.5%)	13 (30.2%)

Categorical variables are summarized with n (%); quantitative variables are summarized with the median (P25, P75) or mean (SD). CRP, C-reactive protein; ICU, intensive care unit; COPD, chronic obstructive pulmonary disease; HIV, human immunodeficiency virus; LOS, length of stay; PaO_2_/FiO_2_, ratio of arterial oxygen partial pressure to fractional inspired oxygen; SAPS, Simplified Acute Physiology Score; VAP, ventilator-associated pneumonia; P25, percentile 25%; P75, percentile 75%; PCT, procalcitonin; PSP, pancreatic stone protein; SD, standard deviation.

**Table 2 biomedicines-11-02676-t002:** Raw and adjusted values of PSP before and after VAP diagnosis.

	Days before VAP			VAP Diagnosis	Days after VAP		
	3	2	1		1	2	3
Raw values (ng/mL), mean (SD)	309.9 (182.2)	225.8 (179.7)	235.2 (165)	254.8 (165.6)	316.5 (201.1)	310.3 (195)	262.3 (173.3)
Adjusted values by means of a mixed-effect model (ng/mL), mean (95% CI)	291.1 (241.5, 341.8)	298.7 (249.6, 348.5)	304.1 (255.1, 353.5)	307.1 (258.4, 356.0)	307.6 (259.7, 355.6)	305.3 (259.1, 351.9)	300.2 (256.0, 344.9)

Adjusted values by means of a mixed-effect model. VAP, ventilator-associated pneumonia; SD, standard deviation; CI, confidence interval.

**Table 3 biomedicines-11-02676-t003:** Adjusted hazard ratios and 95% credible interval for VAP.

		HR (95% CrI) ^a^
Model 1	PSP (normalized to a 0–1 range) value	0.998 (0.87; 1.14)
	PSP (normalized to a 0–1 range) daily change	0.97 (0.64; 1.43)
	PSP (normalized to a 0–1 range) cumulative area	1.04 (0.48; 2.27)
	Infected (vs. Non-infected)	0.35 (0.11; 0.97)
Model 2	PSP (normalized to a 0–1 range) value	1.01 (0.97; 1.06)
	log(CRP) value	0.999 (0.82; 1.22)
	log(PCT) value	0.94 (0.82; 1.07)
	Infected (vs. Non-infected)	0.36 (0.12; 0.99)
Model 3	PSP (normalized to a 0–1 range) daily change	1.15 (0.93; 1.41)
	log(CRP) daily change	0.78 (0.37; 1.59)
	log(PCT) daily change	0.74 (0.40; 1.37)
	Infected (vs. Non-infected)	0.35 (0.11; 0.97)
Model 4	PSP (normalized to a 0–1 range) value	1.01 (0.97; 1.05)
	PSP (normalized to a 0–1 range) daily change	1.15 (0.94; 1.41)
	log(CRP) value	1.004 (0.83; 1.22)
	log(CRP) daily change	0.76 (0.35; 1.58)
	log(PCT) value	0.95 (0.82; 1.08)
	log(PCT) daily change	0.77 (0.40; 1.45)
	Infected (vs. Non-infected)	0.35 (0.11; 0.99)
Model 5	PSP (normalized to a 0–1 range) value	1.07 (0.95; 1.21)
	PSP (normalized to a 0–1 range) daily change	0.93 (0.67; 1.28)
	PSP (normalized to a 0–1 range) cumulative area	0.66 (0.29; 1.39)

^a^ HR calculations. PSP: For both the value and daily change, a difference of 0.172 in the normalized scale for PSP corresponds to a difference of 100 ng/mL in the original scale of PSP, and hence, exp(0.172 × Assoct) gives the corresponding HR for an increase of 100 ng/mL. For the area, HR corresponds to a unit increase in area under the longitudinal profile of the normalized scale for PSP; CRP and PCT: For both the value and daily change, a difference of 0.693 in the log-scale corresponds to a ratio of 2 in the original scale, and hence, exp(0.693 × Assoct) gives the corresponding HR for doubling. For the area, HR corresponds to a unit increase in area under the longitudinal profile of the log-scale for C-RP or PCT. Model 1: the value, daily change, and cumulative area of PSP as associated parameters; Model 2: the value of PSP, CRP, and PCT as associated parameters; Model 3: the daily change of PSP, CRP, and PCT as associated parameters; Model 4: the value and daily change of PSP, CRP, and PCT as associated parameters; Model 5: the value, daily change, and cumulative area of PSP as associated parameters, for patients with non-infection at admission. Models 1–4 are adjusted for the presence of infection at ICU admission. CrI, credible interval; CRP, C-reactive protein; HR, hazard ratio; P, tail probability (2 × min{P(>0),P(<0)}); PCT, procalcitonin; PSP, pancreatic stone protein; SD, standard deviation; VAP, ventilator-associated pneumonia.

## Data Availability

The datasets used and/or analyzed during the current study are available from the corresponding author upon reasonable request.

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
