# Peer review of "Evaluation of the Kinetics of Pancreatic Stone Protein as a Predictor of Ventilator-Associated Pneumonia"

_biomedicines, 2023, doi:10.3390/biomedicines11102676_

Round 1

Reviewer 1 Report

1- What is the significance of early and adequate antibiotic treatment in the context of ventilator-associated pneumonia (VAP)?

2-Were there any limitations mentioned in the abstract regarding the study's methodology or findings?

2-How does Pancreatic Stone Protein (PSP) function as a biomarker, and what is its relevance in the diagnosis of nosocomial sepsis?

3-What was the main hypothesis of the study regarding the use of PSP and its kinetics in the early diagnosis of VAP?

4-Could you provide more details about the methods used in the BioVAP study to evaluate the role of biomarker dynamics in VAP diagnosis?

5-What were the key findings of the study concerning the association between PSP, CRP, and PCT (C-reactive protein and procalcitonin) and the risk of VAP?

6-Were there any significant differences in the predictive value of these biomarkers when considering their daily absolute values or daily changes over time?

7-What is the clinical implication of the conclusion that neither absolute PSP values nor PSP kinetics, alone or in combination with other biomarkers, improved prediction diagnosis accuracy in VAP patients?

8-What potential future research or clinical implications arise from the results of this study, especially considering the challenges in diagnosing VAP early?

9-The following article can be used to improve the discussion parts in the introduction: https://doi.org/10.1016/j.ijbiomac.2022.01.134

Author Response

Sabadell, September 14th, 2023

Dear Editors

We would like to thank you for the opportunity to review our manuscript. Your

comments and those of the reviewers have helped us to introduce important

improvements. We are now re-submitting a revised version, with the changes

highlighted in track changes.

We look forward to hearing from you in due course.

Sincerely yours,

Prof Antonio Artigas and Adrian Ceccato

On behalf of all co-authors. 

Institut d'Investigació i Innovació Parc Taulí 

Parc del Taulí, 1 - 08208 Sabadell, Spain 

Reviewer 1

We would like to thank the reviewer for their comments we are sure improve our manuscript.

What is the significance of early and adequate antibiotic treatment in the context of ventilator-associated pneumonia (VAP)?

Thanks for the comment, We clarified this sentence.

2-Were there any limitations mentioned in the abstract regarding the study's methodology or findings?

Thanks for the comment, all the limitations are explained in the discussion. The word count abstract of the abstract limited the possibility of disclosure limitations.

2-How does Pancreatic Stone Protein (PSP) function as a biomarker, and what is its relevance in the diagnosis of nosocomial sepsis?

Thanks for the comment, PSP function, and previous results with PSP are detailed in the third paragraph of the introduction and the discussion.

3-What was the main hypothesis of the study regarding the use of PSP and its kinetics in the early diagnosis of VAP?

Thanks for the comment, The hypothesis is detailed in the last sentences of the introduction.

4-Could you provide more details about the methods used in the BioVAP study to evaluate the role of biomarker dynamics in VAP diagnosis?

Thanks for the comment, All the methodology was explained in methods. More details could be found in prior published studies or at clinicaltrials.gov

5-What were the key findings of the study concerning the association between PSP, CRP, and PCT (C-reactive protein and procalcitonin) and the risk of VAP?

Thanks for the comment, The results were explained in the results section and discussed in the discussion

6-Were there any significant differences in the predictive value of these biomarkers when considering their daily absolute values or daily changes over time?

Thanks for the comment, The analyses were performed using Bayesian analysis, so the analysis doesn´t give us a p-value, but according to the hazard ratio, PSP does not have enough accuracy to predict VAP by daily absolute values or daily changes

7-What is the clinical implication of the conclusion that neither absolute PSP values nor PSP kinetics, alone or in combination with other biomarkers, improved prediction diagnosis accuracy in VAP patients?

Thanks for the comment, We clarified the conclusions according to the suggestions of both reviewers.

8-What potential future research or clinical implications arise from the results of this study, especially considering the challenges in diagnosing VAP early?

Thanks for the comment, We clarified the conclusions according to the suggestions of both reviewers.

9-The following article can be used to improve the discussion parts in the introduction: https://doi.org/10.1016/j.ijbiomac.2022.01.134

Thanks for the comment, We clarified the introduction according to the reviewer's suggestion.

Reviewer 2 Report

Dear authors,

I was interested to read your article "Evaluation of the kinetics of Pancreatic Stone Protein as a Predictor of Ventilator-Associated Pneumonia. A post-hoc analysis of BioVAP Study"

I think minor improvements would increase the value of the manuscript.

First, no acromins are right into the title, unless explained first.

Second I believe more information about ventilator-associated pneumonia should be detailed not limiting to COVID-19 pandemic.

I believe the methods and the study design of the manuscript are strong and well managed, the statistical is very well conducted.

In the result section maybe some tables and figures from supplementary materials should be introduced into the main body of the manuscript.

Also modify Figure 2, it is too big comparing with the rest.

The fact that only 43 of the 209 patients studied developed VAP should be listed as a limitation of the study, in the limitations listed in the end of discussion section. 

 Discussion section is well conducted but may be improved, conclusions are poor, but sustain the results.

Author Response

Reviewer 2

I was interested to read your article "Evaluation of the Kinetics of Pancreatic Stone Protein as a Predictor of Ventilator-Associated Pneumonia. A post-hoc analysis of BioVAP Study"

I think minor improvements would increase the value of the manuscript.

We would like to thank the reviewer for their comments we are sure to improve our manuscript.

First, no acromins are right into the title, unless explained first.

Thanks for the comment, we change the title according to reviewer comment.

Second I believe more information about ventilator-associated pneumonia should be detailed not limiting to COVID-19 pandemic.

Thanks for the comment, We added more information to the first paragraph about VAP.

I believe the methods and the study design of the manuscript are strong and well managed, the statistical is very well conducted.

Thanks for the comment

In the result section maybe some tables and figures from supplementary materials should be introduced into the main body of the manuscript.

Thanks for the comment, we added a table from supplementary.

Also modify Figure 2, it is too big comparing with the rest.

Thanks for the comment, we modified the fig. 2

The fact that only 43 of the 209 patients studied developed VAP should be listed as a limitation of the study, in the limitations listed in the end of discussion section. 

Thanks for the comment, we added this limitation.

 Discussion section is well conducted but may be improved, conclusions are poor, but sustain the results.

Thanks for the comment, we added a sentence to the conclusions.
